# Neoadjuvant Chemoradiotherapy versus Chemotherapy for Gastroesophageal Junction Adenocarcinoma; Which Is the Optimal Treatment Option?

**DOI:** 10.3390/cancers14235856

**Published:** 2022-11-28

**Authors:** Eric Zandirad, Hugo Teixeira Farinha, Beatriz Barberá-Carbonell, Sandrine Geinoz, Nicolas Demartines, Markus Schäfer, Styliani Mantziari

**Affiliations:** Department of Visceral Surgery, Faculty of Biology and Medicine UNIL, Lausanne University Hospital (CHUV), Rue du Bugnon 46, 1011 Lausanne, Switzerland

**Keywords:** neoadjuvant chemoradiotherapy, neoadjuvant chemotherapy, gastroesophageal junction adenocarcinoma, gastroesophageal junction, neoadjuvant treatment, overall survival

## Abstract

**Simple Summary:**

The optimal neoadjuvant treatment modality for locally advanced gastroesophageal junction (GEJ) adenocarcinoma is still debated. Although chemoradiotherapy is set as the standard of treatment for squamous cell cancer, offering high rates of complete clinical and histologic response, the optimal treatment for adenocarcinoma remains a matter of debate. This study retrospectively compared 94 patients with locally advanced adenocarcinoma of the esophago-gastric junction treated with neoadjuvant chemoradiotherapy (*n* = 27) versus chemotherapy (*n* = 67) followed by curative surgery. Chemoradiotherapy offered better histological response of the primary tumor, but no benefit in terms of negative resection margins or long-term survival or recurrence. Patients undergoing chemoradiation were shown to have higher rates of cardiovascular complications after surgery. Based on these findings, the added benefit of external beam radiation in the neoadjuvant treatment of locally advanced esophageal adenocarcinoma remains unclear.

**Abstract:**

*Background*: Locally advanced gastroesophageal junction adenocarcinoma (GEJ) is treated with either perioperative chemotherapy (CT) or preoperative radiochemotherapy (RCT) followed by surgery. The aim of this study was to compare pathologic response and long-term outcomes in junction adenocarcinoma treated with neoadjuvant RCT versus CT. *Methods*: All patients with locally advanced GEJ adenocarcinoma treated with neoadjuvant treatment (NAT) followed by surgery between 2009 and 2018 were retrospectively analyzed. *Results*: A total of 94 patients were included, 67 (71.2%) RCT and 27 (28.8%) CT. Complete pathologic response was more frequent in RCT patients (13.4% vs. 7.4%, *p* = 0.009) with a trend to better lymph node control (ypN0) (55.2% vs. 33.3%; *p* = 0.057). RCT offered no benefit in R0 resection (66.7% vs. 72.1% CT, *p* = 0.628) and was related to higher postoperative cardiovascular complications (35.8% vs. 11.1%; *p* = 0.017). Long-term overall and disease-free survival were similar (5-year OS 61.1% RCT vs. 75.7% CT, *p* = 0.259; 5-year DFS 33.5% RCT vs. 22.8% CT; *p* = 0.763). NAT type was neither independently associated with pathologic response nor long-term survival. *Discussion*: Patients with locally advanced GEJ adenocarcinoma treated with RCT had more postoperative cardiovascular complications but higher rates of complete pathologic response and a trend to superior locoregional lymph node control. This did not translate in a survival or recurrence benefit.

## 1. Introduction

Esophageal cancer is the sixth most common cancer worldwide, with over 600,000 newly diagnosed cases per year [1,2]. Recent advances, and especially multimodal neoadjuvant treatment for locally advanced disease, have improved overall prognosis, with median survival increasing from 22 to 56 months during the past 30 years in surgical series [3]. Although radiochemotherapy (RCT) is recommended for the more radiosensitive SCC subtype, the optimal modality for AC is not so clearly defined [2]. In these patients, RCT and chemotherapy (CT) are both indicated, often used interchangeably without specific selection criteria [2,4,5].

In 2010, two randomized trials suggested better local disease control with RCT compared to CT; however, they were both prematurely closed due to low accrual, not demonstrating differences in survival [6,7]. The Swedish NeoRes trial [8,9] favored neoadjuvant RCT compared to CT in terms of R0 resection and histologic response, although SCC and AC histology were considered jointly, somewhat blurring the conclusions. Markar et al. assessed a large series of junctional AC patients [10]; they reported notably low rates of complete response after CT (5%) compared to RCT (26.7%), in a series treated in the pre-CROSS and pre-FLOT era. Two retrospective large-scale studies recently published comparative results of CROSS and FLOT regimens in AC patients, with conflicting results for histologic response, and no survival benefit of one modality over the other [11,12]. Finally, tree randomized trials (ESOPEC [13], Neo-AEGIS [14], and RACE [15]) are ongoing to compare the current standards of treatment. ESOPEC and Neo-AEGIS trials have meanwhile completed recruitment and are in their follow-up phases with results awaited within the next few years. 

Following the publication of the Checkmate-577 trial, targeted PD-1/PD-L1 checkpoint inhibitors are rapidly gaining acceptance in the adjuvant setting in patients who received RCT without obtaining a complete response [16]. Such evidence does not yet exist after CT. Finally, the watch-and-wait strategy with surgery on-demand is suggested as a treatment option not only for SCC, but also for AC patients with clinical complete response after RCT [17]. Thus, different treatment strategies may be offered to AC patients depending on the type of preoperative regimen, highlighting the importance of wise and evidence-based use preoperative treatment modalities. To this day, the question of RCT versus CT remains open for esophageal AC, as the exact patient benefit from one modality over the other remains unclear.

The aim of the present study was to compare the rates of complete pathologic response, long-term survival, and recurrence in a series of consecutive patients with locally advanced AC of the esophagus or gastroesophageal junction, treated with neoadjuvant RCT or CT and oncologic surgery. 

## 2. Materials and Methods

### 2.1. Study Design

All patients operated on at our tertiary referral center for locally advanced adenocarcinoma of the esophagus or gastroesophageal junction between 1 January 2009 and 31 December 2018 were retrospectively assessed. Comparative analyses were performed between those who received neoadjuvant chemotherapy (CT group) versus radiochemotherapy (RCT group).

### 2.2. Inclusion and Exclusion Criteria

Inclusion criteria were age >18 years with locally advanced adenocarcinoma of the distal esophagus and gastroesophageal junction (Siewert I–II), who underwent NAT followed by surgery with curative intent after discussion in the multidisciplinary tumor board. Tumoral staging was made with systematic CT-scan and PET-CT for tumoral localization and distant metastases. Endosonography was conducted to assess the T and N stage. All perioperative CT regimens (e.g., 5FU-cisplatin, ECF/ECX, 5FU-leucovorin-oxaliplatin-docetaxel-FLOT) were eligible, whereas the neoadjuvant RCT group included patients who received radiotherapy between 41 and 50.4 Gy. Emergency surgery, all other histological types, and Siewert III gastric tumors were excluded from analysis, as were patients with documented refusal to participate in clinical research. In addition, patients treated with upfront surgery without NAT as well as those who underwent salvage surgery after exclusive radiotherapy were excluded from the study.

### 2.3. Ethics and Reporting

The present study was conducted according to the Declaration of Helsinki ethical standards. All included patients provided written consent, and the study was approved by the local ethics committee (CER-VD Protocol No. 2019-02368). 

### 2.4. Study Definitions and Outcome Measures

Baseline and preoperative global health status were assessed with the WHO performance status [18] and American Society of Anesthesiologists (ASA) class, respectively. Postoperative complications were graded according to the validated 5-scale Clavien classification, and major morbidity was defined as a grade >IIIAc [19]. The 8th version of the UICC/TNM system was used to assess the baseline and histologic tumor stage [20] whereas R0 status was defined as a <1 mm microscopic margin from the tumor [21]. The primary outcome was the rate of pathological complete response (pCR) in RCT and CT patients; this was defined as the complete sterilization of the primary tumor (TRG1 grade according to the Mandard regression system) [22]. Secondary outcomes were overall and disease-free survival, estimated in months after the beginning of NAT, rates of R0 resection, postoperative complications, and treatment toxicity. 

### 2.5. Statistical Analysis

Categorical variables are presented as absolute frequency (%) and continuous as the mean (standard deviation—SD). Comparison of the categorical variables was performed with the χ2 or Fisher’s test as appropriate, while continuous variables were compared with the non-parametric Mann–Whitney-U test. For time-to-event outcomes such as overall survival (OS) and disease-free survival (DFS), the Kaplan–Meier method and log-rank test was used. All comparisons were performed in an intention-to-treat analysis. All tests were two-sided and statistical significance threshold was set at *p* < 0.05. Postoperative follow-up was assessed with the reverse Kaplan–Meier method. Statistical analyses were carried out using SPSS (version 23.0, Chicago, IL, USA) and RStudio (Version 1.1.383, Boston, MA, USA) software.

## 3. Results

### 3.1. Baseline Characteristics

Overall, 94 patients were included in the present study: 67 in the RCT and 27 in the CT group (Figure 1). Baseline demographics and tumor stage did not present significant differences between the groups. On baseline endoscopy, the Z-line as well as the proximal margin of the tumor were located higher up the esophagus in RCT compared to the CT patients (Table 1).

### 3.2. Treatment Details, Postoperative Outcomes (Table 2)

The majority of patients in the RCT group (82%) received a platin/taxol chemotherapy regimen, whereas in the CT group perioperative EOX (51.8%) and FLOT (33.3%) were the predominant modalities (*p* < 0.001). One patient from the chemotherapy group was switched to a RCT strategy during treatment due to CT-related toxicity (severe ileitis during EOX chemotherapy). NAT modification due to severe toxicity was performed in *n* = 9 (33.3%) patients in the CT versus *n* = 11 (16.4%) in the RCT group. Transthoracic esophagectomy according to Lewis was the technique in all patients. Surgery occurred after a median of 50.3 [IQR 14.9] days in the RCT group, and 43.3 days [IQR 14.8] in the CT group (*p* = 0.052). 

Postoperative complications were similar in both groups, with major morbidity (Clavien ≥IIIA) occurring in 47.8% RCT vs. 40.7% CT patients (*p* = 0.998). Cardiovascular complications were significantly higher in the RCT than in the CT group (35.8% vs. 11.1%, *p* = 0.017). RCT and CT patients had the following ICU stay median oof 8.8 vs. 3.2 days, *p* = 0.086 and overall hospital stay was similar. 

**Table 2 cancers-14-05856-t002:** Surgical characteristics and postoperative outcomes for all patients.

Surgical and Postop. Variables	All Patients *N* = 94	RCT *N* = 67	CT *N* = 27	*p*-Value
Surgical approach				0.086
Lewis Santy (%)	86 (91.5)	64 (95.5)	22 (81.5)	
Transhiatal (%)	8 (8.5)	3 (4.5)	5 (18.5)	
Laparoscopy (%)	84 (89.3)	61 (91)	23 (85.1)	0.404
Thoracosopy (%)	45 (47.8)	29 (43.2)	16 (59.2)	0.077
Operative time (min)	300.4 (71.9)	306.8 (77)	284.5(55.2)	0.122
NAT-Surgery interval (days)	48.4 (15.1)	50.3 (14.9)	43.3 (14.8)	0.052
Clavien–Dindo grade [19] (%)				0.988
0	31 (33)	21 (31.3)	10 (37)	
I	2 (2.1)	1 (1.5)	1 (3.7)	
II	18 (19.1)	13 (19.4)	5 (18.5)	
IIIA	6 (6.3)	4 (6)	2 (7.4)	
IIIB	10 (10.6)	7 (10.4)	3 (11.1)	
IV	22 (23.4)	17 (25.3)	5 (18.5)	
V	5 (5.3)	4 (6)	1 (3.7)	
Pulmonary complications (%)	48 (51)	36 (53.7)	12 (44.4)	0.376
Cardiovascular complications (%)	27 (28.7)	24 (35.8)	3 (11.1)	0.017
Anastomotic leakage (%)	27 (28.7)	20 (29.8)	7 (26)	0.704
Length of stay (days)	28.3 (31.9)	30.1 (34.9)	23.7 (22.1)	0.287
Length of ICU stay (days)	7.1 (20.2)	8.8 (23.5)	3.2 (6.5)	0.086

Categorical variables are expressed as *N* (%), and continuous variables as median [IQR]. NAT = neoadjuvant treatment; ICU = intensive Care Unit.

### 3.3. Histological Analysis, Tumor Response to Treatment 

Complete pathologic response (pCR, TRG1) was achieved more frequently in the RCT group (13.4% vs. 7.4% in CT patients), while complete absence of histologic response (TRG 5) was only seen in the CT group (11.1% vs. 0% in RCT patients, *p* = 0.009) (Table 3). The R0 resection rates were similar in both groups (73.1% RCT versus 66.6% in CT patients, *p* = 0.628).

There was a trend to higher rates of lymph node sterilization (ypN0 stage) in the RCT group (55.2% vs. 33.3% in CT patients; *p* = 0.057). Microscopic lymphatic invasion was observed in 51.8% CT and 26.8% RCT patients (*p* = 0.041), whereas perineural invasion was observed in 48.1% CT versus 26.8% RCT patients, *p* = 0.096). The RCT group had a lower number of lymph nodes retrieved in the surgical specimen (median 21.6 [IQR 7.6] versus 31.3 [IQR 13] in CT patients, *p* < 0.001). Simple logistic regression identified no preoperative variables significantly associated with pCR, thus multivariate analysis was not performed (Appendix A).

### 3.4. Long-Term Survival and Recurrence Patterns

Median follow-up was 30 months (95%CI 21.3–38.8) (33 months for the RCT group and 21 months for the CT group) and median survival was 71 months (95%CI 64.1–77.9) for all patients. No significant difference was observed between the RCT and CT group in terms of disease free survival (DFS); as 3-year DFS survival was 35.3% in RCT versus 24.3% for CT patients (*p* = and overall survival (OS); 3-year OS was 78.6% in RCT and 75.7% in CT patients (*p* = 0.259) (Figure 2). In multivariate analysis, only cN2 status (HR 4.0, 95% CI 1.2–13.2) and major postoperative complications (HR 1.2, 95%CI 1.2–8.1) were significantly associated with poor OS (Appendix A).

Median DFS for all patients was 21 months (95%CI 16.7–25.3); 18 months (95%CI 13.6–22.5) in RCT, and 23 months (95%CI 8.9–37.1) in CT patients. Respective 3-year DFS was 35.9% and 22.8% (*p* = 0.763). (Figure 3). Multivariate analysis revealed only cN2 status to be significantly associated with poor DFS (Appendix A). 

In this series, 24.4% of all patients presented an early tumor recurrence, within 12 months from surgery (25.3% in the RCT vs. 22.2% in the CT group; *p* = 0.748). Isolated locoregional recurrence was present in 3% RCT vs. 3.7% CT patients, whereas 19.4% RCT patients and 18.5% CT patients presented both simultaneous locoregional and disseminated recurrence (*p* = 0.976).

## 4. Discussion

In the present series of gastroesophageal junction AC patients, RCT offered better pathological response rates compared to CT alone. However, RCT patients did not have an improved overall or disease-free survival, and the recurrence patterns were similar.

As the objectives of neoadjuvant treatment are local downsizing and downstaging of the tumor, the elimination of micrometastases, and the prevention of distant progression, pathological response to treatment is often used as a surrogate of treatment efficacy. In the present study, 13.7% versus 7.4% patients after RCT and CT, respectively, had a pathologic complete response (pCR), whereas all patients with no response at all belonged to the CT group. These results are in line with previous studies, reporting significantly higher pCR after RCT (18–28%) compared to CT (5–10%) [9,10,11,23]. Conversely, Donlon et al., in a recent large-scale study comparing the FLOT and CROSS regimens, reported similar pCR rates (15% vs. 13%, *p* = 0.36) [12]. Nevertheless, can the primary tumor’s response to treatment really be considered as a reliable surrogate for patient outcomes? 

Locoregional lymph node control is just as important in order to assess the efficacy of the neoadjuvant treatment. In the present study, despite similar baseline clinical stage, we observed a trend to higher lymph node sterilization in the RCT group, with significantly less microscopic lymphovascular invasions. Surgical technique was extensive two-field lymphadenectomy in all patients, with no systematic differences after RCT or CT, but RCT patients were still found to have a lower number of lymph nodes retrieved in the surgical specimen (median 21.6 versus 31.3). Lower lymph node yields have already been reported in a previously irradiated field compared to surgery or chemotherapy alone [10,24,25]. Locoregional fibrosis and the loss of lymph node structure after radiation may explain this finding [24]. Although the benefit from extensive surgical lymphadenectomy may be more pronounced for patients without neoadjuvant treatment [26], the quest for adequate lymph node dissection should not be abandoned in RCT patients hoping that radiation will compensate for suboptimal lymph node yield. Lutfi et al. recently showed that even in cases of complete pathologic response of the primary tumor after neoadjuvant treatment, adequate lymphadenectomy (>=15 nodes) remained associated with improved survival [27].

Unfortunately, up to this day, the superior locoregional disease control offered by RCT has failed to translate into a tangible recurrence or survival benefit for patients. Only a minority of patients, similar between RCT and CT patients (3% in each group), presented locoregional recurrence in our series, as suggested by previous studies [9,10]. Goense et al. did report lower 3-year rates of locoregional recurrence in RCT patients (16% versus 37%, *p* = 0.022), but still no overall survival benefit [23]. The long-term results of the POET trial [28] reported a trend to better overall survival after RCT, although our data, as long as previous long-term studies, do not confirm this finding [7,8,9,10,12,23]. Recently, patients with pCR after CT were reported to have better recurrence-free survival than those with pCR after RCT [29]. Moreover, it has to be kept in mind that radiation is no longer an option in the case of locoregional recurrence in patients already treated with RCT preoperatively. Thus, the better local control offered by RCT may be followed by some potentially serious disadvantages in the long-term, even if complete histologic response was seen on the specimen.

Another important aspect is the treatment-related toxicity associated with each modality. Potent CT regimens are often plagued by severe (>grade II) toxicity, reported in 25% patients in the landmark FLOT trial [4] and confirmed in 33% of CT patients in the present series. In contrast, postoperative morbidity seems increased after RCT. In the present study, cardiovascular complications appeared more often in the RCT group, as previous authors have suggested [23]. Other studies have reported an increase (3-fold) in anastomotic leak rates [10], higher rates of respiratory failure [12], and more severe (Clavien > IIIa) complications in RCT compared to CT patients [30]. The long-term morbidity of radiation also has to be considered. Cardiopulmonary toxicity and fibrosis, although minimized with modern radiation techniques, is an evolving process over several months post-treatment. In long-term follow-up of NeoRes patients, chronic cough and persistent impairment in overall health-related QoL, even 5 years after surgery, were significantly more frequent after RCT [31]. In the present study, we also observed a trend to a higher length stay and of ICU stay.

The present study has some limitations that need to be considered. First, the relatively small sample size, as this was a single-center series of a subset of esophageal cancer patients with a minimum 3-year postoperative follow-up. The present study is also a non-randomized design with its inherent bias. However, the high quality of the prospectively recorded data as well as the standardization in surgical oncologic treatment and histologic analysis of specimens compensate for the limited patient number. Smaller series with good-quality data may be of high scientific value, not only to reflect the local practice and results in different parts of the world, but also to provide basis for future systematic reviews and meta-analyses on the subject. Of note, different chemo and radiation regimens were used in the present series. Despite the heterogeneity this might induce, it reflects a real-world situation where RCT regimen choice remains variable among centers end even oncologists at the same center.

## 5. Conclusions

In conclusion, the present study suggests that patients with locally advanced AC of the gastroesophageal junction present more favorable histologic response of primary tumor and locoregional lymph node control when treated with neoadjuvant RCT compared to CT. However, this benefit did not translate in a survival or recurrence benefit for RCT patients, who even seemed to suffer more postoperative cardiovascular adverse events.

## Figures and Tables

**Figure 1 cancers-14-05856-f001:**
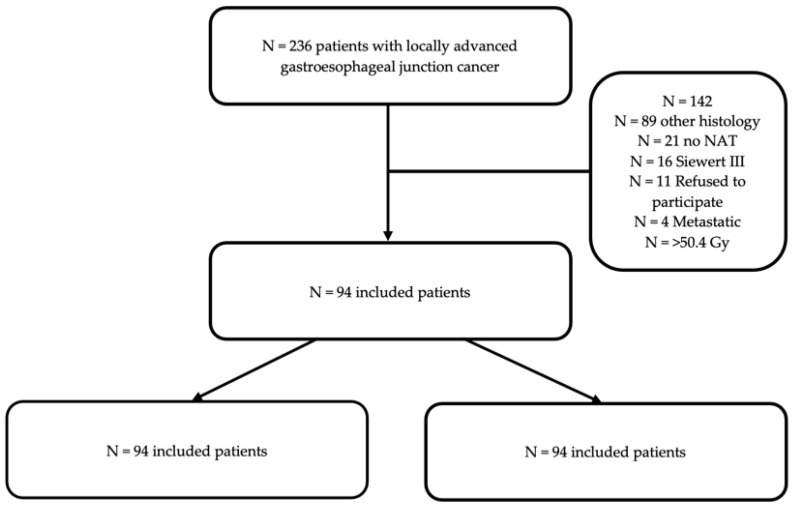
Flowchart of the study description of the patient inclusion in the study group for analysis.

**Figure 2 cancers-14-05856-f002:**
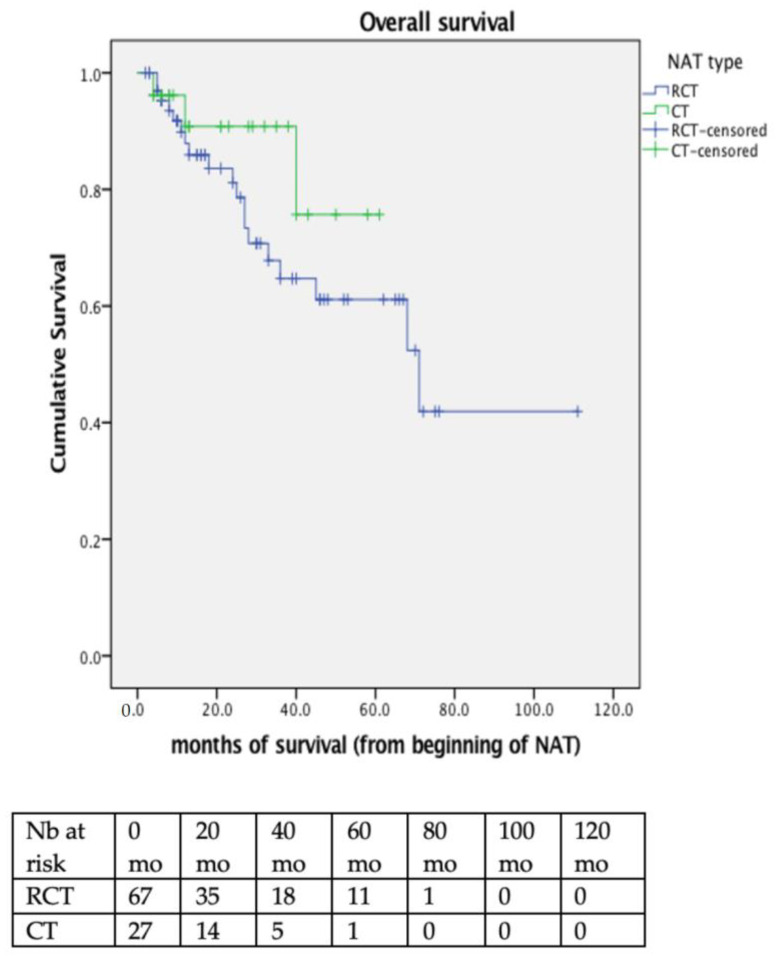
Overall survival (OS) for all patients, according to their treatment group. RCT patients had a median OS of 71 months (95%CI 37.5–104.4) whereas median OS was not reached in the CT group. 3-year OS was 78.6% in RCT and 75.7% in CT patients (*p* = 0.259). NAT = neoadjuvant treatment; RCT = radio-chemotherapy; CT = chemotherapy.

**Figure 3 cancers-14-05856-f003:**
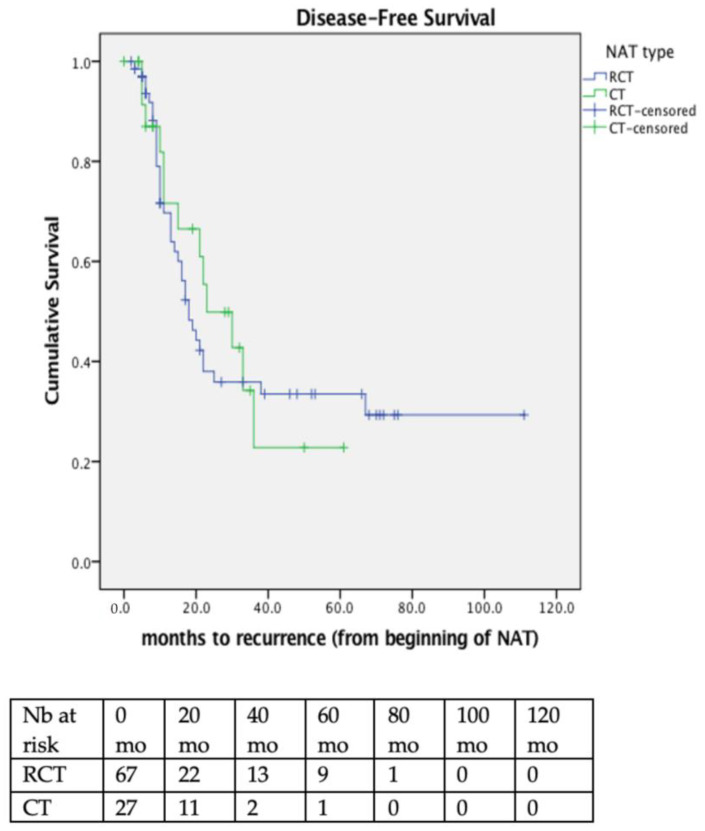
Disease-free survival (DFS) for all patients, according to their treatment group. Median DFS was 18 months (95%CI 13.6–22.5) in RCT and 23 months (95%CI 8.9–37.1) in CT patients. Respective 3-year DFS was 35.9% and 22.8% (*p* = 0.763). NAT = Neoadjuvant treatment; RCT = radio-chemotherapy; CT = chemotherapy.

**Table 1 cancers-14-05856-t001:** Demographic and preoperative characteristics of the study population.

Demographic Variables	All Patients *N* = 94	RCT *N* = 67 (71%)	CT *N* = 27 (29%)	*p*-Value
Male gender (%)	79 (84)	57 (85.1)	22 (81.5)	0.667
ASA class				0.833
1–2 (%)	64 (68.0)	46 (68.6)	18 (66.6)	
3–4 (%)	28 (29.7)	19 (28.3)	9 (33.3)	
Unknown	2	-	-	
Age, years	62 [9.7]	62.1 [10.1]	61.9 [8.7]	0.927
BMI, kg/m^2^	25.2 [4.0]	24.7 [3.8]	26.4 [4.0]	0.064
Weight loss at baseline (kg)	6.7 (5.6)	6.5 (5.5)	7.6 (5.9)	0.569
Weight loss during NAT (kg)	3.0 (6.7)	3.6 (6.3)	1.2 (7.3)	0.219
Baseline WHO status				0.150
0–1 (%)	71 (75.5)	53 (79.1)	18 (66.7)	
2–3 (%)	23 (24.4)	14 (20.9)	9 (33.3)	
GERD (%)	54 (57.4)	36 (53.7)	18 (66.6)	0.251
Distance to the Z-line (cm) *	38.5 (4.5)	37.9 (4.9)	40.1 (2.6)	0.022
Superior tumor margin (cm) *	34.5 (3.5)	34.0 (3.6)	36 (3.0)	0.013
Inferior tumor margin (cm) *	39.5 (3.3)	39.1 (3.1)	40.8 (3.7)	0.074
cT stage				0.897
2 (%)	7 (7.2)	5 (7.4)	2 (7.4)	
3–4 (%)	76 (80.8)	56 (83.5)	20 (74)	
pTx	11	-	-	
cN stage				0.984
0 (%)	20 (21.3)	15 (22.4)	5 (18.5)	
			14 (51.8)	
			3 (11.1)	
SUVmax (g/L)	14 [7.1]	14.5 [7.5]	12.7 [5.6]	0.233
Radiotherapy dose				<0.001
41.4 Gy (%)	25 (26.5)	24 (35.8)	1 ** (3.7)	
45 Gy (%)	17 (18.0)	17 (25.3)	0 (0)	
50.4 Gy (%)	22 (23.4)	22 (32.8)	0 (0)	
Chemotherapy regimen				<0.001
5FU-platin (%)	5 (5.3)	3 (4.4)	2 (7.4)	
EOX (%)	17 (18.0)	3 (4.4)	14 (51.8)	
FLOT (%)	10 (10.6)	1 (1.5)	9 (33.3)	
Platin-Taxol (%)	56(59.5)	55(82)	1(3.7)	
Others (%)	5 (5.3)	5 (7.4)	0 (0)	
Unknown	1	0	1	
Treatment modification due to toxicity (%)	20 (21.2)	11 (16.4)	9 (33.3)	0.076

Categorical variables are expressed as *N* (%), and continuous variables as median [IQR]. ASA = American Society of Anesthetists score; BMI = body mass index; NAT= neoadjuvant treatment; WHO = World Health Organization; GERD = gastroesophageal reflux disease; GEJ = gastro-esophageal junction; SUVmax = baseline maximal standardized uptake value on PET-CT imaging. * Median distance from the superior dental arch (cm). ** One patient had to switch from CT to RCT group due to chemo-induced ileitis. Analyses were performed as intention-to-treat.

**Table 3 cancers-14-05856-t003:** Histopathologic characteristics of the surgical specimen.

Histopathologic Variables	All Patients *N* = 94	RCT *N* = 67	CT *N* = 27	*p*-Value
ypT stage (%)				0.858
0	11 (11.7)	9 (13.4)	2 (7.4)	
1	13 (13.8)	8 (12)	5 (18.5)	
2	10 (10.6)	8 (12)	2 (7.4)	
3	57 (60.3)	40 (59.7)	17 (63)	
4	3 (3.2)	2 (3)	1 (3.7)	
ypN stage (%)				0.057
0	46 (49)	37 (55.2)	9 (33.3)	
1	21 (22.3)	14 (20.9)	7 (26)	
2	17 (18)	12 (18)	5 (18.5)	
3	10 (10.6)	4 (6)	6 (22.2)	
R1 resection (%)	24 (25.5)	16 (23.8)	8 (29.6)	0.628
Circumferential resection margin (mm)	3.0 (6.7)	2.4 (3.1)	4.2 (11.2)	0.492
Tumor regression grade (TRG) [22] (%)				0.009
1	11 (11.7)	9 (13.4)	2 (7.4)	
2	24 (25.5)	19 (28.3)	5 (18.5)	
3	20 (21.2)	18 (26.8)	2 (7.4)	
4	33 (35.1)	19 (28.3)	14 (51.8)	
5	3 (3.2)	0 (0)	2 (11.1)	
Lymphovascular invasion (L1) (%)	32 (34)	18 (26.8)	14 (51.8)	0.041
Microvascular invasion (V1) (%)	27 (28.7)	17 (25.3)	10 (37)	0.305
Perineural invasion (Pn1) (%)	31 (32.0)	18 (26.8)	13 (48.1)	0.096
Signet-ring histology (%)	16 (17)	10 (15)	6 (22.2)	0.410
HER2 (+) status (%)	8 (8.5)	6 (9)	2 (7.4)	0.472
Positive lymph nodes (%)	2.3 [4.2]	1.7 [2.6]	4.0 [6.5]	0.083
Harvested lymph nodes (%)	24.4 [10.5]	21.6 [7.6]	31.3 [13]	0.001
Barrett’s metaplasia (%)	34 (36.1)	27 (40.3)	7 (26)	0.256

Categorical variables are expressed as *N* (%), and continuous variables as median [IQR]. TRG = tumor regression grade; HER2 = Human Epidermal Growth Factor Receptor 2.

## Data Availability

Data are maintained in this article.

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
