# Peer review of "Neoadjuvant Chemoradiotherapy versus Chemotherapy for Gastroesophageal Junction Adenocarcinoma; Which Is the Optimal Treatment Option?"

_cancers, 2022, doi:10.3390/cancers14235856_

Round 1

Reviewer 1 Report

The authors present a single-centre retrospective study of RCT vs RT as a NAT for Siewert I-II EAC. The study is well-designed, the statistical analysis is sound and the conclusions generally supported by the data. The article is written in good English and well presented. 

In my opinion, a few minor points should be addressed. 

1. Materials and methods. The exact staging procedures should be detailed (CT, PET, Endosonography?)

2. Numbers in Table 1 should be verified and possibly explained. First, the header of the table should specify numbers (%). Second, ASA-classification is shown for 92 patients only. Third, cT and cN stages are indicated for 83 patients only. If there are no data for the remaining patients, those should be indicated as pTx and pNx. 

3. The non-significant tendency towards longer hospital and ICU stays could be discussed, especially as the non-significant tendency for higher lymph node sterilisation is mentioned. 

4. Numbers in table 2 should be checked. Specifically, chemotherapy regimen is indicated for 93 patients only, the missing patient being in the CT group. 

Author Response

Reviewer 1:

1. Materials and methods. The exact staging procedures should be detailed (CT, PET, Endosonography?)

Thank you for this remark, we add this information.

2. Numbers in Table 1 should be verified and possibly explained. First, the header of the table should specify numbers (%). Second, ASA-classification is shown for 92 patients only. Third, cT and cN stages are indicated for 83 patients only. If there are no data for the remaining patients, those should be indicated as pTx and pNx. 

Thank you for these important remarks, we add all modifications in Table 1 + Missing values

3. The non-significant tendency towards longer hospital and ICU stays could be discussed, especially as the non-significant tendency for higher lymph node sterilization is mentioned. 

We added a sentence.

4. Numbers in table 2 should be checked. Specifically, chemotherapy regimen is indicated for 93 patients only, the missing patient being in the CT group. 

We corrected it. For the remaining patient we did not have the information.

Reviewer 2 Report

The manuscript addresses an interesting and clinically relevant topic and is generally well written. The non-randomized design is a clear limitation of the study, which needs to be considered in critically appreciating the results.

I have a few suggestions for improvement of the mauscript:

Introduction: 

- With regard to ongoing trials comparing neoadjuvant chemotherapy and radiochemotherapy for AEG, also the currently recruiting RACE trial (Lorenzen S, Biederstädt A, Ronellenfitsch U, Reißfelder C, Mönig S, Wenz F, Pauligk C, Walker M, Al-Batran SE, Haller B, Hofheinz RD. RACE-trial: neoadjuvant radiochemotherapy versus chemotherapy for patients with locally advanced, potentially resectable adenocarcinoma of the gastroesophageal junction - a randomized phase III joint study of the AIO, ARO and DGAV. BMC Cancer. 2020 Sep 15;20(1):886. doi: 10.1186/s12885-020-07388-x.) should be mentioned. The ESOPEC and NeoAegis trials have meanwhile completed recruitment and are in their follow-up phases, which should also be mentioned.

Methods: 

- I would not use the term "primary endpoint", but rather "primary outcome", for a retrospective study.

- Do you have data on the median/mean follow-up in the overall study population and the two defined groups? These should be mentioned in the text.

Results:

- The number of patients at risk at different time points should be shown in the graphs with the survival curves.

- Can you provide results on how many patients in the respective group received postoperative chemotherapy or radiochemotherapy?

Discussion:

- I disagree with the statement that "the primary objective of neoadjuvant treatment is local downsizing and downstaging of the tumor". While this is beyound doubt one objective, elimination of micrometastases and thus prevention of distant progression should be considered an equally important aim of neoadjuvant therapy.

- In the methodological discussion, where the limitations of the present study are addressed, you need to mention the non-randomized design with its inherent bias. 

Author Response

Introduction: 

- With regard to ongoing trials comparing neoadjuvant chemotherapy and radiochemotherapy for AEG, also the currently recruiting RACE trial (Lorenzen S, Biederstädt A, Ronellenfitsch U, Reißfelder C, Mönig S, Wenz F, Pauligk C, Walker M, Al-Batran SE, Haller B, Hofheinz RD. RACE-trial: neoadjuvant radiochemotherapy versus chemotherapy for patients with locally advanced, potentially resectable adenocarcinoma of the gastroesophageal junction - a randomized phase III joint study of the AIO, ARO and DGAV. BMC Cancer. 2020 Sep 15;20(1):886. doi: 10.1186/s12885-020-07388-x.) should be mentioned. The ESOPEC and NeoAegis trials have meanwhile completed recruitment and are in their follow-up phases, which should also be mentioned.

Thank you for this important remark, we add this reference and adapt the introduction.

Methods: 

- I would not use the term "primary endpoint", but rather "primary outcome", for a retrospective study.

Adapted.

- Do you have data on the median/mean follow-up in the overall study population and the two defined groups? These should be mentioned in the text.

The median follow-up is 30 months for the whole study population, 33 months for the RCT group and 21 months for the CT group. (Calculated with the reverse Kaplan meier method). We added this information in results.

Results:

- The number of patients at risk at different time points should be shown in the graphs with the survival curves.

Thank you for the comment, they have been added in respective tables with the number at risk for each group.

- Can you provide results on how many patients in the respective group received postoperative chemotherapy or radiochemotherapy?

Unfortunately, we do not have this information for all patients.

Discussion:

- I disagree with the statement that "the primary objective of neoadjuvant treatment is local downsizing and downstaging of the tumor". While this is beyound doubt one objective, elimination of micrometastases and thus prevention of distant progression should be considered an equally important aim of neoadjuvant therapy.

Thank you for this remark, we reformulated the sentence as proposed.

- In the methodological discussion, where the limitations of the present study are addressed, you need to mention the non-randomized design with its inherent bias.

We add this point on discussion session.

Reviewer 3 Report

The authors have presented the results of a comparison of two retrospective cohorts of patients with GEG/distal esophageal adenocarcinoma treated with neoadjuvant chemoradiotherapy or perioperative chemotherapy.

As the authors themselves have said in the introduction and in the discussion of this paper this issue is a ever-present topic of research, as results of randomised phase III trials that should give us a definitive answer on this topic are still ongoing (ESOPEC) or we only have some preliminary results (Neo-Aegis). On top of that, I believe that retrospective data based on everyday clinical practice is still useful even when the results of large randomised studies are pending, as it reflects the everyday challenges of this group of patients that requires accurate multidisciplinary management.

The methods of this analysis have been explained in detail and are mostly reproducible. The results have been presented also in a clear fashion and are easy to understand.

I have only a few issues with this paper that I would like the authors to manage:

1. Even if the small sample size may pose a limitation, retrospective comparison between two different cohorts of patients treated with 2 different "standards" of treatment, should be managed by taking into account some matching procedure (p.e. propensity score matching). I would suggest the authors to keep the results that they have reported in the unmatched population and to perform matching procedure between patients treated with chemotherapy (lower number of patients) vs those treated with chemoradiotherapy. 

2. In table 1 (patients' demographics) number of patients does not match: in particular, as per cT and cN part of the table it is stated that only 83/94 patients have cT2-4 and that 83/94 patients have cN0-3. Should I guess that 11/94 (12%) patients have cTx and cNx status or different? I believe that this is a relevant issue as the results of this paper (and other previous related articles that have been cited by the authors) have suggested that chemoradiotherapy has higher response rate and lower rate of positive lymphnodal involvement after lymphnodal harvest: inadequate lymphnodal staging before the beginning of neoadjuvant treatment should be taken into account. In the case that these 11 patients have inadequate cT and cN staging this should be specified at least in the tables.

3. Authors have described different profiles of toxicities of both chemoradiotherapy and perioperative chemotherapy; I believe that the authors should also add up how many patients were able to complete the scheduled treatment. In particular, I would like to know how many patients in the peri-operative chemotherapy group were able to complete the scheduled (usually 2-3 months) post-operative chemotherapy. We know that after surgery the number of patients who are able to complete post-operative chemotherapy is quite small but it is also the group of patients who have the highest chance of being cured: it is thus necessary to know how many patients in the peri-operative chemotherapy were also able to receive "adjuvant" treatment in addition to "neo-adjuvant" therapy.

4. Perioperative FLOT chemotherapy was used in a minority of patients in the chemotherapy cohort. Because of that we can't fully discriminate whether FLOT perioperative chemotherapy is comparable to neoadjuvant radiochemotherapy. As patients were collected in a long timeframe (2009-2018) I would like to know whether there has been a change in surgeon/oncologist decision whether to commit a patients to radiochemotherapy opposed to perioperative chemotherapy based on the year when the patient was assessed (as I expect that more patients in the chemotherapy perioperative treatment are relatively more "recent" compared to patients in the radiochemotherapy group).

5. Survival calculations have yielded some strange results: authors stated that median follow-up is 30 months whereas median OS is 71 months. I believe that these results are partly explained by the fact that the follow-up time is immature for overall survival calculations, particularly in the perioperative chemotherapy subgroup (mOS for this group of patients is not yet reached). I would focus more on DFS analysis as 30 months median follow-up time should be more than enough to make assumptions on this.

I believe that these issues should be assessed before the paper can be accepted for publication.

Author Response

Reviewer 3:

  1. Even if the small sample size may pose a limitation, retrospective comparison between two different cohorts of patients treated with 2 different "standards" of treatment, should be managed by taking into account some matching procedure (p.e. propensity score matching). I would suggest the authors to keep the results that they have reported in the unmatched population and to perform matching procedure between patients treated with chemotherapy (lower number of patients) vs those treated with chemoradiotherapy. 

As the baseline demographics are comparable, given the small size of the group and without notable differences, a matching procedure seems unreasonable to us in this context.

  1. In table 1 (patients' demographics) number of patients does not match: in particular, as per cT and cN part of the table it is stated that only 83/94 patients have cT2-4 and that 83/94 patients have cN0-3. Should I guess that 11/94 (12%) patients have cTx and cNx status or different? I believe that this is a relevant issue as the results of this paper (and other previous related articles that have been cited by the authors) have suggested that chemoradiotherapy has higher response rate and lower rate of positive lymphnodal involvement after lymphnodal harvest: inadequate lymphnodal staging before the beginning of neoadjuvant treatment should be taken into account. In the case that these 11 patients have inadequate cT and cN staging this should be specified at least in the tables.

Thank you for these remarks, reviewer 1 also raised this point. We adapted the table.

  1. Authors have described different profiles of toxicities of both chemoradiotherapy and perioperative chemotherapy; I believe that the authors should also add up how many patients were able to complete the scheduled treatment. In particular, I would like to know how many patients in the peri-operative chemotherapy group were able to complete the scheduled (usually 2-3 months) post-operative chemotherapy. We know that after surgery the number of patients who are able to complete post-operative chemotherapy is quite small but it is also the group of patients who have the highest chance of being cured: it is thus necessary to know how many patients in the peri-operative chemotherapy were also able to receive "adjuvant" treatment in addition to "neo-adjuvant" therapy.

Thank you for this remark. This point is in fact very important, unfortunately due to the retrospective study with inherent missing values, we do not have this information

  1. Perioperative FLOT chemotherapy was used in a minority of patients in the chemotherapy cohort. Because of that we can't fully discriminate whether FLOT perioperative chemotherapy is comparable to neoadjuvant radiochemotherapy. As patients were collected in a long timeframe (2009-2018) I would like to know whether there has been a change in surgeon/oncologist decision whether to commit a patients to radiochemotherapy opposed to perioperative chemotherapy based on the year when the patient was assessed (as I expect that more patients in the chemotherapy perioperative treatment are relatively more "recent" compared to patients in the radiochemotherapy group).

Unfortunately for the same reason as before, it is very difficult or even impossible to return to each folder to retrieve this precise information.

  1. Survival calculations have yielded some strange results: authors stated that median follow-up is 30 months whereas median OS is 71 months. I believe that these results are partly explained by the fact that the follow-up time is immature for overall survival calculations, particularly in the perioperative chemotherapy subgroup (mOS for this group of patients is not yet reached). I would focus more on DFS analysis as 30 months median follow-up time should be more than enough to make assumptions on this.

This is a right point, effectively these results could be explained by immature follow up time. DFS analysis was reported on results session, (line 191 – 193) we add median DFS and 3 years DFS (%)